# Definition, Assessment, and Management of Vitamin D Inadequacy: Suggestions, Recommendations, and Warnings from the Italian Society for Osteoporosis, Mineral Metabolism and Bone Diseases (SIOMMMS)

**DOI:** 10.3390/nu14194148

**Published:** 2022-10-06

**Authors:** Francesco Bertoldo, Luisella Cianferotti, Marco Di Monaco, Alberto Falchetti, Angelo Fassio, Davide Gatti, Luigi Gennari, Sandro Giannini, Giuseppe Girasole, Stefano Gonnelli, Nazzarena Malavolta, Salvatore Minisola, Mario Pedrazzoni, Domenico Rendina, Maurizio Rossini, Iacopo Chiodini

**Affiliations:** 1Emergency Medicine, Department of Medicine, University of Verona, 37129 Verona, Italy; 2Bone Metabolic Diseases Unit, Department of Experimental, Clinical and Biomedical Sciences, University of Florence, 50121 Florence, Italy; 3Osteoporosis Research Center, Fondazione Opera San Camillo, Presidio di Torino, 10131 Torino, Italy; 4Laboratory of Experimental Clinical Research on Bone Metabolism, Istituto Auxologico Italiano, Istituto di Ricovero e Cura a Carattere Scientifico (IRCCS), 20145 Milan, Italy; 5Rheumatology Unit, University of Verona, Policlinico GB Rossi, 37134 Verona, Italy; 6Department of Medicine Surgery and Neuroscience, University of Siena, 53100 Siena, Italy; 7Clinica Medica 1, Department of Medicine, University of Padova, 35122 Padova, Italy; 8Rheumatology Department, “La Colletta” Hospital, ASL 3 Genovese, 16011 Arenzano, Italy; 9Casa di Cura Madre Fortunata Toniolo, and Centri Medici Dyadea, 40141 Bologna, Italy; 10U.O.C. Medicina Interna A, Malattie Metaboliche dell’Osso Ambulatorio Osteoporosi e Osteopatie Fragilizzanti, Sapienza University of Rome, 00185 Rome, Italy; 11Department of Medicine and Surgery, University of Parma, 43121 Parma, Italy; 12Department of Clinical Medicine, Surgery Federico II University, 80138 Naples, Italy; 13Unit for Bone Metabolism Diseases and Diabetes, Istituto Auxologico Italiano, Istituto di Ricovero e Cura a Carattere Scientifico (IRCCS), 20133 Milan, Italy; 14Department of Medical Biotechnology and Translational Medicine, University of Milan, 20122 Milan, Italy

**Keywords:** vitamin D, bone metabolism, osteoporosis, bone fragility, chronic diseases

## Abstract

In the recent years, both the prescriptions of serum 25(OH)D levels assay, and vitamin D supplementation are constantly increasing, as well as the costs to be incurred relating to these specific aspects. As in many other countries, the risk of vitamin D deficiency is particularly high in Italy, as recently confirmed by cohort studies in the general population as well as in patients with metabolic bone disorder. Results confirmed the North-South gradient of vitamin D levels described among European countries, despite the wide use of supplements. Although vitamin D supplementation is also recommended by the Italian Medicine Agency for patients at risk for fragility fracture or for initiating osteoporotic medication, the therapeutic gap for osteoporosis in Italy is very high. There is a consistent proportion of osteoporotic patients not receiving specific therapy for osteoporosis following a fragility fracture, with a poor adherence to the recommendations provided by national guidelines and position paper documents. The failure or inadequate supplementation with vitamin D in patients on antiresorptive or anabolic treatment for osteoporosis is thought to further amplify the problem and exposes patients to a high risk of re-fracture and mortality. Therefore, it is important that attention to its possible clinical consequences must be given. Thus, in light of new evidence from the literature, the SIOMMMS board felt the need to revise and update, by a GRADE/PICO system approach, its previous original recommendations about the definition, prevention, and treatment of vitamin D deficiency in adults, released in 2011. Several key points have been here addressed, such as the definition of the vitamin D status: normality values and optimal values; who are the subjects considered at risk of hypovitaminosis D; opportunity or not of performing the biochemical assessment of serum 25(OH)D levels in general population and in subjects at risk of hypovitaminosis D; the need or not to evaluate baseline serum 25(OH)D in candidate subjects for pharmacological treatment for osteoporosis; how and whether to supplement vitamin D subjects with hypovitaminosis D or candidates for pharmacological treatment with bone active agents, and the general population; how and whether to supplement vitamin D in chronic kidney disease and/or chronic liver diseases or under treatment with drugs interfering with hepatic metabolism; and finally, if vitamin D may have toxic effects in the subject in need of supplementation.

## 1. Introduction

Vitamin D plays a relevant role in maintaining a healthy mineralized skeleton and in preventing rickets and osteomalacia [1]. Humans get vitamin D (as either vitamin D2 [ergocalciferol] or vitamin D3 [cholecalciferol]) from sunlight, diet, or supplements. After entering circulation, its activity depends on liver hydroxylation to 25-hydroxyvitamin D [25(OH)D] and then to 1,25-dihydroxyvitamin D [1,25(OH)2D], the active hormone. The latter hydroxylation mainly occurs in the kidney, but also activated macrophages, parathyroid glands, microglia, breasts, the colon, and keratinocytes retain this function [2].

The measurement of serum 25(OH)D, which includes 25(OH)D2 and 25(OH)D3 forms, is used in clinical practice to assess the so-called vitamin D status and is interpreted as an expression of the body “vitamin D reserve”. In fact, the 25(OH)D form is relatively stable in serum with a half-life of 2–3 weeks, while its activated form, 1,25(OH)2D, has a half-life of about 15 h [3].

In 2011, the Institute of Medicine (IOM) defined the threshold 25(OH)D levels for deficiency, insufficiency and sufficiency as <12 ng/mL (30 nmol/L), between 12 and 20 ng/mL (30 and 50 nmol/L) and between 20 and 30 ng/mL (50 and 75 nmol/L), respectively [4]. However, scientific societies such as the Endocrine Society, the National Osteoporosis Foundation, and the International Osteoporosis Foundation, suggested that sufficiency levels should be based on values > 30 ng/mL [5,6,7].

In the 2016 report, the Italian Society for Osteoporosis, Mineral Metabolism and Bone Diseases (SIOMMMS) suggested optimal 25(OH)D range values [8]. There is good evidence and a broad agreement that 25(OH)D < 12 ng/mL (30 nmol/L) levels are associated with rickets, osteomalacia and secondary hyperparathyroidism [9]; therefore it is unanimously agreed that such values constitute a real condition of vitamin D deficiency [4,5,8].

The risk of vitamin D deficiency (either considering the 10 ng/mL or the 20 ng/mL threshold) is particularly high in Italy, as recently confirmed by cohort studies in the general population as well as in patients with metabolic bone disorders [10], thus mirroring the North-South gradient of vitamin D levels described among European countries, despite the wide use of supplements [11]. This implies an important attention to its possible clinical consequences and the need for action deriving from the wide prevalence of hypovitaminosis D and the cost-efficacy concerns related to the measurement of serum vitamin D levels and vitamin D supplementation [12,13].

Vitamin D depletion may reduce the protection against fractures provided by several drugs widely used for the treatment of osteoporotic patients in clinical practice [14]. For these reasons, vitamin D supplementation is also recommended by the Italian Medicine Agency (Agenzia Italiana del Farmaco, AIFA) for patients at risk of fragility fracture or initiating osteoporotic medication [15]. However, the therapeutic gap for osteoporosis in Italy is very high [16] and a consistent proportion of osteoporotic patients did not receive specific therapy for osteoporosis following a fragility fracture, demonstrating poor adherence to the recommendations provided by national guidelines and position paper documents [17]. Failure or inadequate supplementation with vitamin D in patients on antiresorptive or anabolic treatment for osteoporosis further amplifies the problem and exposes patients to a high risk of re-fracture and mortality [18].

Based on the above premises, and in the light of the new evidences from the literature, the SIOMMMS board felt the need to revise and update its original recommendations about the definition, prevention and treatment of vitamin D deficiency in adults, released in 2011 [19].

## 2. Purposes

A task force composed of expert representatives of the SIOMMMS was established in order to provide clinical guidelines on the diagnosis and management of hypovitaminosis D with the following main purposes: (a) to improve and standardize the “clinical practice”; (b) to offer the patient the indications for “best care”, to be followed uniformly at the national level; and (c) to guarantee an evidence-based reference for national and regional institutions, for regulatory organizations and payers. This position statement is primarily intended for use by clinicians.

Specifically, we address the following key points and sub-points:Definition of the vitamin D status: normality values and optimal valuesWho are the subjects at risk of hypovitaminosis D?Should the biochemical assessment of serum 25(OH)D levels be conducted in the general population?Should the biochemical assessment of serum 25(OH)D levels be conducted in the population at risk of hypovitaminosis D?
Is there any direct evidence that basal 25(OH)D levels represent an essential parameter for prescribing vitamin D supplementation?Is there any evidence, in a population at risk of hypovitaminosis D, that a basal 25(OH)D measurement may contribute to preventing potential toxicity due to vitamin D supplementation?Is the baseline 25(OH)D measurement cost-effective, in a population at risk of hypovitaminosis D?
Is baseline serum 25(OH)D testing necessary in candidate subjects for pharmacological treatment of osteoporosis?How should vitamin D be supplemented in subjects with hypovitaminosis D or candidates for pharmacological treatment with bone active agents?Should the general population be supplemented with vitamin D?How should the patient with chronic kidney disease (CKD) be supplemented with vitamin D?How should the patient with chronic liver diseases or under treatment with drugs interfering with hepatic metabolism be supplemented with vitamin D?Might vitamin D have toxic effects in the subject who need supplementation?

## 3. Methods

The SIOMMMS Executive Council selected representative national experts in the field to participate with the working board. The board included the SIOMMMS President and members representing the Executive Council itself. An experienced academic endocrinologist (FB) was selected to chair the working group. All potential conflicts of interests of participating authors were declared before manuscript drafting.

A computerized literature search was performed in PubMed; Cochrane Central; Cochrane Database of Systematic Reviews; and the Health Technology Assessments; search limited to English-language articles on the adult population. (last update: 10 January 2022) using the terms “target population”, “purpose of supplementation”, “vitamin D status”, “recommendation of vitamin D supplementation”, “thresholds for deficiency or insufficiency”, “screening advice”, “monitoring”, “sun exposure”, “method to obtain vitamin D”, “vitamin D testing”, “25(OH)D2”, “25(OH)D32”, “risk of low vitamin D”, “fragility fracture”, ”fall”, “osteomalacia”, “primary hyperparathyroidism”, “secondary hyperparathyroidism”, “kidney failure”.

Depending on the clinical question, the more appropriate study designs to answer the question were considered. Editorials, position papers, comments, letters, narrative reviews, or case reports were not included.

The Grading of Recommendations, Assessment, Development, and Evaluation (GRADE) system was adopted for the present position paper [20,21,22,23].

According to GRADE, evidence was revised based on 5 dimensions (risk of bias, imprecision, inconsistency, indirectness, publication bias) and categorized into four quality levels (high, moderate, low, or very low), while recommendations were classified as strong (“recommendations”) or weak (“suggestions”), on the basis of the quality of supporting evidence and level of agreement between the panel members. The panel then proceeded to evaluate the benefit/damage ratio related to the intervention in question (vote) and participated in the vote on the strength of the recommendation relating to the clinical question posed. The vote was expressed according to the majority.

All authors contributed to the writing of the manuscript, and the final draft statement was agreed to by all authors.

### 3.1. Definition of the Vitamin D Status: Normality Values and Optimal Values

Levels of serum 25(OH)D vary throughout the different periods of life, depending on the season, latitude, degree of sunlight exposure, phototype and body mass index (BMI). Moreover, at present, significant drawbacks are encountered both in the field of research and in clinical practice deriving from the analytical variability in the dosage of serum vitamin D levels.

Today, the assessment of serum 25(OH)D levels is mostly performed using immuno-chemiluminescence methods, which are characterized by an intra-assay and inter-laboratory variability of 10–20%. Therefore, there is an urgent need for standardization/harmonization of dosages both for a correct interpretation of clinical studies and for clinical practice [24]. In this respect, liquid chromatography tandem mass spectrometry is currently considered the most accurate and precise method not only for research but also in clinical practice.

The definition of “normality” levels and vitamin D deficiency is a heavily debated topic. While there is unanimous agreement that values of 25(OH)D < 10 ng (<25 nmol/L) represent a condition of severe deficiency, a consensus for what can be considered “normality” does not exist. This aspect has a major impact on both the epidemiology of hypovitaminosis D and the clinical practice, with consequent repercussions on the prescription of vitamin D supplements.

As first, the problem of defining the adequate serum level of 25(OH)D requires some clarification regarding the terms “normal value” and “optimal value”. With the former, we generally refer to a statistically determined level defined as an average ± 2 standard deviations (DS) of the values detected in a given population. This information is of particular relevance to researchers and institutions dealing with studies concerning the general population. In this respect, there may be different normality values for 25(OH)D depending on the different geographical areas, age groups, and seasonality. In contrast, the “optimal” or “desirable” level is defined as the value, which has proved effective in achieving the prevention of a disease and/or of a related adverse event (e.g., fractures), based on the evidence provided by observational and intervention studies designed ad hoc. Therefore, scientific societies express a “recommended serum level” of 25(OH)D based on the profile of the patient and the outcome to be pursued.

While there is a general consensus on the threshold of 10–12 ng/mL to define a condition of “vitamin D deficiency” (associated with rickets, osteomalacia and secondary hyperparathyroidism) the definition of “sufficiency” values in the general population remains controversial. In order to determine a possible cut-off for 25(OH)D sufficiency, the association between vitamin D levels and different outcomes were considered, including the level of suppression of parathyroid hormone (PTH), intestinal calcium absorption or other aspects related to skeletal health and, in particular, fracture risk. This might in part explain the different range of values concerning the “sufficient” or “optimal” 25(OH)D levels reported in the literature for the different skeletal outcomes, highlighting the impossibility to define a precise cut-off level. In this respect, the information about the sufficient serum 25(OH)D values for extra-skeletal outcomes is even more inconsistent [25]. Indeed, the attempt to define as the sufficient threshold the 25(OH)D value that normalizes PTH does not seem entirely convincing, since marked differences have been reported for this outcome across the different studies, with a wide range of 25(OH)D levels, oscillating between 12 ng/mL (30 nmol/L) and 36 ng/mL (90 nmol/L) [26]. Furthermore, the interaction curve does not show a true plateau point for PTH suppression at the 30 ng/mL (75 nmol/L) threshold of 25(OH)D, as described by Pepe and coworkers [27], and most importantly it considerably varies depending on the different age groups and calcium intake [28].

Thus, for the definition of a sufficient 25(OH)D level in the general population, the association between vitamin D deficiency and fractures has been considered as the most relevant indicator by members of this task force. There is a good but not unanimous consensus on the association between serum values of 25(OH)D less than 20 ng/mL (50 nmol/L) and increased risk of fracture [29]. A recent meta-analysis reported that for values less than 20 ng/mL (50 nmol/L) there is a 40% increase in the risk of femoral fracture for each SD decrease of 25(OH)D [30]. Similarly, another meta-analysis of prospective cohort studies showed that the risk of fracture is linearly reduced to a value of 25(OH)D of approximately 24 ng/mL (60 nmol/L), while for values higher than this threshold the fracture risk would no longer decrease [31].

On the contrary, there is no evidence that serum 25(OH)D values above 20 ng/mL (50 nmol/L) may lead to relevant advantages for skeletal health (i.e., bone mineral density (BMD) or fractures) in the general population. In a large randomized controlled clinical trial (RCT) performed on healthy adults, high doses of cholecalciferol or vitamin D3 (equal to 100,000/IU per month) for about 4 years did not lead to any advantage in terms of risk of falls and fractures, compared to the placebo arm of the study. Since 80% of the population studied had a basal 25(OH)D > 25 ng/mL (60 nmol/L), these results indicate that this value can be considered sufficient and adequate for the general population and that there are no specific reasons or advantages to introduce vitamin D supplement in these subjects [32].

A recent meta-analysis, performed on the musculoskeletal effects of vitamin D supplementation, partly supports, and confirms this concept. In fact, there was no significant effect on BMD and fractures, but 55% of the studies included in the meta-analysis recruited patients with baseline values > 20 ng/mL (50 nmol/L) and only 6% of them enrolled patients with values < 10 ng/mL (25 nmol/L). Again, the data suggest that vitamin D supplementation in subjects with 25(OH)D values ≥ 20 ng/mL (50 nmol/L) does not lead to any advantage and that, therefore, this level can be considered adequate in the general population [33].

However, it is important to point out that the proposed threshold for “sufficient levels” of 25(OH)D ≥ 20 ng/mL (50 nmol/L), in which vitamin D supplementation does not seem to add any particular advantage, refers to the “normal”, healthy population, and therefore not at high risk of hypovitaminosis D, as indicated in Table 1

These “normal” subjects often represent the majority of individuals included in prospective population studies and RCT, in which cholecalciferol supplementation has shown no significant clinical outcome to date. In fact, in a large meta-analysis of 9 RCTs including healthy adult subjects (selected because they were not at risks of osteoporosis, fractures, falls, use of osteopenia drugs, etc.), supplementation with cholecalciferol at doses from 700 to 3000 IU/day had no effect on fractures, mortality or morbidity [34].

Likewise, the definition of a correct target of 25(OH)D levels for the categories of subjects in which supplementation is appropriate, is fundamental in order to avoid an excessive use of supplements that will not provide any real benefit [35]. This has recently led to the inclusion, in a non-critical manner, of vitamin D among the *overused* drugs [36].

There is also a general, evidence-based, agreement that in subjects at risk of hypovitaminosis D (Table 2) and in subjects treated with anti-resorptive or anabolic drugs (mainly osteoporotic patients) the supplementation of vitamin D is highly recommended.

In the same RCTs meta-analysis described above, in which the global effect of vitamin D supplements (with or without calcium) on fractures appears to be non-significant, a significant advantage in terms of reduction of the fracture risk was particularly observed in the subgroup of institutionalized subjects or individuals with previous fractures [31]. These concepts are also endorsed by the European Society for Clinical and Economic Aspects of Osteoporosis, Osteoarthritis and Musculoskeletal Diseases (ESCEO) and by the International Osteoporosis Foundation (IOF) [37]. Likewise, in a different meta-analysis of RCTs in women older than 60 years, having as an outcome the reduction of the risk of femoral and non-vertebral fractures, a significant effect of vitamin D supplementation was obtained in those who reached serum 25(OH)D values above 30 ng/mL (75 nmol/L), with a 20% and 18% reduction for non-vertebral and femoral fractures, respectively. Indeed, patients treated with bisphosphonates with a mean 25(OH)D ≥ 33 ng/mL had a roughly 4.5-fold greater odds of a favorable response than patients with lower levels [38,39].

After the revision of the available information, the panel concluded that the upper recommended level defining “optimal” 25(OH)D status in the general population could be set at 50 ng/mL (125 nmol/L). This threshold was based upon data showing that, particularly concerning falls and mortality, there might be a U-shaped curve defining the beneficial effects of serum 25(OH)D, as if beyond this level there may be the resumption of negative clinical outcomes. However, a recent study, where a standardization of 25(OH)D dosages was reached, clearly showed how the correlation curve between vitamin D levels and mortality is not properly U-shaped, but it progressively flattens (i.e., J-shape) with a plateau at values around 18–20 ng/mL (45–50 nmol/L) [40]. This also suggests that reaching 25(OH)D levels higher than 30 ng/mL is not of particular relevance for the general population, but it is relatively safe.

In summary, notwithstanding the above limitations, the definition of optimal 25(OH)D levels is of major relevance not only for epidemiological estimates but also for clinical practice. Thus, we propose that in the general population, including healthy elderly individuals, a threshold value of 25(OH)D ≥ 20 ng/mL (50 nmol/L) should be considered as adequate and should not require any supplementation. Conversely, in patients with osteoporosis or other disorders of bone metabolism, especially when treated with *bone modifying agents*, as well as in subjects at risk for hypovitaminosis D (as indicated in Table 2 and addressed in the following chapter), a value of ≥30 ng/mL (75 nmol/L) can be suggested as “optimal” (Table 1).

### 3.2. Who Are the Subjects at Risk of Hypovitaminosis D?

This clinical question implies as primary step the identification of those factors or clinical conditions that increase the risk of hypovitaminosis D (differentiating them from the general population). Mainly, this risk is clinically identifiable with a series of well-defined and universally accepted conditions [8,26,41]. Advanced age is commonly considered a risk factor, however most of the epidemiological data does not support the presence of lower 25(OH)D levels in the non-institutionalized elderly population, compared to those found in adulthood [42]. This is most likely due to the widespread use of vitamin D supplements in the over 65 population, while these levels are lower in the younger population [43,44]. The relationship between the risk of falls and low vitamin D levels is generally accepted, but elderly subjects with a history of falls are not always included in the list of subjects at risk for hypovitaminosis D [45,46,47,48]. In an RCT performed on the effect of vitamin D supplements on falls, 409 Finnish women—with ages ranging between 70 and 80 years, with at least 1 fall during the previous year, and without supplementation of vitamin D—had sufficient baseline values of 25(OH)D (26.2–27.8 ng/mL) [49].

Concerning dietary habits, the vegan diet may be included in the list of risk factors for hypovitaminosis D, while there is no general consensus on the vegetarian diet [47,50,51].

Among eating disorders, anorexia nervosa should be considered a condition at risk of hypovitaminosis D [52,53]. Other subject categories at risk of severe hypovitaminosis also include patients with solid tumors (mainly breast, prostate, colon and lung), obesity or type 2 diabetes mellitus [54,55,56,57,58]. A list of the most relevant conditions at risk of hypovitaminosis D identified by this panel is summarized in Table 2.

**Table 2 nutrients-14-04148-t002:** Population/condition at risk of hypovitaminosis D.

Old people (≥75 years)
Institutionalized subjects or conditions associated with inadequate solar exposure
Obesity
Pregnancy and breast-feeding
Metabolic bone diseases and other skeletal disorders
Vegan diet
Anorexia nervosa
Chronic renal failure
Cancer (in particular breast, prostate, and colon)
Type 2 diabetes mellitus
Intestinal malabsorption and bariatric surgery
Drugs that interfere with the absorption or hepatic metabolism of vitamin D (antiepileptics, glucocorticoids, antiviral AIDS, antifungal agents, cholestyramine)
Cystic fibrosis

### 3.3. Should the Biochemical Assessment of Serum 25(OH)D Levels Be Conducted in the General Population?

The measurement of 25(OH)D, widely available although qualitatively questionable if not standardized according to the DEQAS (Vitamin D External Quality Assessment Scheme) and VDSP (Vitamin D Standardization Program) systems, has drastically increased worldwide in the last decade [59]. This has clearly increased health expenditure for both public and private systems [60,61], thus underlining the necessity to avoid unnecessary measurements. This imposes primarily the selection, and the appropriate identification of patients to be subjected to the assessment of 25(OH)D levels [62].

Importantly, no study published to date evaluated the efficacy and safety of vitamin D supplementation according to a randomization performed on screening versus non-screening for vitamin D deficiency [60,61,62,63,64], as recently concluded by an updated report and systematic review for the US Preventive Services Task Force [65]. Therefore, at this stage, we do not recommend the extensive screening for hypovitaminosis D in the adult general population, since there is still no evidence that this procedure and the consequent treatment of vitamin D deficient cases, would represent a cost-effective procedure [64,66] (Table 3).

### 3.4. Should the Biochemical Assessment of Serum 25(OH)D Levels Be Conducted in the Population at Risk of Hypovitaminosis D?

In order to fulfill this issue, the following main sub-questions were addressed:(A)Is there any direct evidence that basal 25(OH)D levels represent an essential parameter for prescribing vitamin D supplementation?

Although the majority guidelines consider the measurement of serum 25(OH)D levels as highly recommendable, at least in subjects defined at risk of hypovitaminosis D, there is no direct evidence supporting a clear advantage in performing an assessment of the basal vitamin D status [5,8,26,41].

Theoretically, the goal of 25(OH)D testing should be to facilitate the normalization of 25(OH)D levels, with potential skeletal, muscular or extra-skeletal benefits [5,8,17,26,41,45,67]. However, the baseline 25(OH)D level does not appear to influence efficacy outcomes such as fractures or falls. The US Preventive Service Task Force (USPSTF) in a systematic review of studies on vitamin D supplementation for the prevention of falls and fractures in adult subjects (aged over 18 years), did not indicate any benefit of assessing basal vitamin D levels on treatment outcomes (exclusion criteria were individuals with levels of 25(OH)D less than 30 ng/mL (75 nmol/L) [65].

Available information shows that the basal assay of vitamin D has little or no influence at all on the 25(OH)D levels reached with supplementation. Binkley et al. [68], in a RCT performed on 62 postmenopausal women randomized to receive 1800 IU/day of cholecalciferol or placebo for 4 months, found no correlation between baseline and endpoint 25(OH)D values. A recent meta-analysis of 10 RCTs aimed at evaluating the association between levels of 25(OH) and fat mass in both observational studies and RCTs did not identify baseline 25(OH)D levels, together with age and duration of supplementation, as a source of heterogeneity of RCTs [69]. Likewise, a systematic review including 144 cohorts from 94 independent studies showed that the basal 25(OH)D levels would explain only a minimal proportion of the response to vitamin D supplementation (1.9%), while 34.5% could be attributed to the dose per kg of body weight, 19% to the given metabolite of vitamin D, 3.7% to age and 2.4% to the use of calcium supplements [70]. Different outcomes emerged from a study by Shab-Bidar and collaborators [71], who assessed the determinants of serum 25(OH)D when patients are given vitamin D supplements, using a random-effect model in a meta-regression analysis of 33 supplementation studies. In this study, baseline 25(OH)D levels (including a subgroup analysis of patients with values below 20 ng/mL [50 nmol/L]), trial duration, age, and supplementation dosage (>800 IU/day) were all independent predictors of vitamin D response. However, a significant heterogeneity between studies was observed [71]. Other studies suggested that it is possible to predict vitamin D deficiency with simple algorithms without dosing 25(OH)D, using indirect data only. Merlijn et al. [72] demonstrated in a population of 2689 women over the age of 65 that they could predict levels of 25(OH)D lower than 12 and 16 ng/mL (30 nmol/L and 40 nmol/L). Sohl et al. [73], as part of the Longitudinal Aging Study Amsterdam, also validated a prediction algorithm for 25(OH)D levels < 12 ng/mL (30 nmol/L) and <20 ng/mL (50 nmol/L) in 1509 women over 70 years, using 13 risk predictors with AUC 0.78 and 0.80, respectively. Nabak et al. [74] recruited 609 postmenopausal women with a mean age of 61 (SD 6 years), of whom 113 (19%) had serum 25(OH)D < 20 ng/mL, to determine whether a questionnaire can identify individuals with vitamin D insufficiency. In logistic regression models, black race, BMI, suntan within the past year, sun exposure in the past 3 months, sunscreen use, and supplemental vitamin D intake were the most useful questions to identify vitamin D insufficiency. From these six items, a composite score of ≤2.25 demonstrated ≥89% sensitivity but ≤35% specificity [74].

Importantly, as previously outlined, 25(OH)D levels persistently below 10–12 ng/mL (25–30 nmol/L) generally support the diagnosis of vitamin D-dependent osteomalacia (especially if associated with high values of alkaline phosphatase). Thus, in the suspicion of signs/symptoms referable to osteomalacia, it is advisable to quantify serum 25(OH)D levels [75,76], mainly for supporting the diagnosis and for the differential diagnosis.

(B)Is there any evidence that, in a population at risk of hypovitaminosis D, the basal 25(OH)D measurement may contribute to prevent potential toxicity?

There are no direct data exploring whether baseline assessment of 25(OH)D is a predictor for the risk of toxicity during supplementation. At the same time, many studies showed that supplementation with high doses of vitamin D are safe also in subjects with 25(OH)D levels in the range of sufficiency (>20 ng/mL, 50 nmol/L). In institutionalized women with values over 20 ng/mL (50 nmol/mL), doses of 500,000 IU cholecalciferol *in bolus* followed by 50,000 IU/month did not lead to hypercalcemia or other adverse events [77]. Similarly, supplementation with cholecalciferol 100,000 IU/month in institutionalized women with a mean baseline 25(OH)D level of 22 ng/mL (55 nmol/L) did not result in toxicity [78]. Interestingly, in many studies, supplementation in subjects with sufficient baseline values of 25(OH)D, near or above 23 ng/mL (57.5 nmol/L), led to a lower increase in 25(OH)D than in those subjects with basal levels below 20 ng/mL (50 nmol/L) [79,80,81,82,83].

(C)Is the baseline 25(OH)D measurement cost-effective in a population at risk of hypovitaminosis D?

There are no data on the cost/effectiveness of determining the basal 25(OH)D level in the categories of subjects at risk for hypovitaminosis D, as defined in Table 2. The National Institute for Health and Care Excellence (NICE) guidelines indicate that, although the dosage of 25(OH)D should be performed only in the population at risk, further studies are required to define the cost/benefit ratio of the dosage itself [84]. The Ontario Health Technology Advisory Committee (OTACH) 2010 report also concluded that the clinical benefit, the social, ethical and economic values of the dosage in subjects at risk of hypovitaminosis D are not yet well defined, even if the approach is very feasible [26]. These concepts are raised in Table 4.

### 3.5. Is the Baseline Serum 25(OH)D Testing Necessary in Candidate Subjects for Pharmacological Treatment for Osteoporosis? 

As stated in point 1 and Table 1, in patients with osteoporosis or other metabolic bone diseases, vitamin D status should be optimal (>30 ng/mL or 75 nmol/L), in order to support the effectiveness of antiresorptive or anabolic agents and to reduce the incidence of adverse events associated with anti-absorptive agents (i.e., parenteral bisphosphonates, denosumab), such as hypocalcemia [85].

In fact, the efficacy of bisphosphonates, teriparatide and denosumab in preventing osteoporotic fractures has been mostly demonstrated when associated with vitamin D supplements. Indeed, most of the RCTs of antiresorptive or osteoanabolic drugs in postmenopausal women and in men with osteoporosis and/or fragility fractures were conducted in association with pre-established and fixed doses of vitamin D. In most of these RCTs, supplementation was given independently of the basal 25(OH)D levels, while in others it was administered in subjects with values lower than 30 ng/mL (75 nmol/L) and greater than 12 ng/mL (30 nmol). Moreover, failure to associate vitamin D supplementation with bone active agents may significantly reduce the anti-fracture efficacy of these drugs, worsening the cost to benefit ratio of the drug itself [14,18] and representing a potential risk of a further fracture [86]. Therefore, it is necessary to ensure an adequate vitamin D supplementation in association with any specific therapy for osteoporosis, also in order to ensure that the circulating 25(OH)D levels reach at least the “optimal” value of 30 ng/mL (75 nmol/L) [87].

However, since supplementation with vitamin D and, eventually, calcium (in order to optimize daily calcium intake) is a mandatory recommendation in this setting, for the primary and secondary prevention of fragility fractures (and particularly in subjects starting an anti-fracture therapy) the determination of basal 25(OH)D is not considered necessary.

Indeed, the verification of the achievement of ”optimal” 25(OH)D levels before the start of anti-fracture therapy, where standard doses are used, might have some benefit for the effectiveness of the therapy itself and eventually prevent possible complications (i.e., hypocalcemia and, in case of intravenous regimens of bisphosphonates acute phase reaction) [88,89]. However, the cost-effectiveness of this approach remains to be demonstrated. Similar indications can be drawn in case of bone antiresorptive or anabolic treatment for other skeletal diseases [5,85,90,91] (Table 5).

### 3.6. How Should Vitamin D Be Supplemented in Subjects with Hypovitaminosis D or Candidate to Pharmacological Treatment with Bone Active Agents?

For vitamin D supplementation, regardless of the threshold to be reached, it is difficult to suggest common rules to be adopted for all individuals. As first, the dose-response relationship is widely variable among individuals, being influenced by many factors, and particularly by intestinal absorption efficiency, the amount of body fat, and gene polymorphisms (i.e., the *DBP* gene encoding the vitamin D-binding protein, or the *CYPR1* gene encoding the enzyme 25-hydroxylase). Moreover, age, concomitant diseases (i.e., malabsorption syndromes) and type of compound used for supplementation are further variables that could significantly affect the achievement of a normal vitamin D status [92,93,94]. Based on these premises, the Endocrine Society suggested the administration of a daily cholecalciferol dose of 1500–2000 IU per day, to reach the desired value of 30 ng/mL (75 nmol/L) [41]. However, the correction of severe hypovitaminosis D or overt osteomalacia with the doses generally recommended for maintenance (800–2000 IU/day) requires several months [14,71]. A quick repletion is needed in the presence of symptoms of vitamin D deficiency (i.e., osteomalacia) and/or values of 25(OH)D < 10 ng/mL (25 nmol/L), as well as in patients eligible for treatment with potent antiresorptives with rapid and prolonged effect on bone resorption, such as intravenous bisphosphonates and denosumab. In this setting, the use of higher loading dosages allows the serum levels of 25(OH)D to be normalized within a few weeks [95,96,97,98,99,100,101,102,103].

To reach a 25(OH)D level of at least 30 ng/mL (75 nmol/L) doses of 3000 to 10,000 daily IU cholecalciferol (in average 5000 IU/day) (or equivalent weekly) has been suggested over 1–2 months [5,8,104]. In a recent comparative, randomized, open label study in adults with low vitamin D levels (<20 ng/mL), daily (10,000 IU/day for 8 weeks followed by 1000 UI/day for 4 weeks), weekly (50,000 IU/week for 12 weeks) or every other week (100,000 IU every other week for 12 weeks) cholecalciferol dosing schedules were all effective and enabled 25(OH)D levels above 30 ng/mL to be reached in most cases (93%), without adverse effects [99,103]. This study showed also that the daily doses were more efficient than boluses, at the same cumulative dose [103]. In addition, it is interesting to note that several studies showed daily administration regiments to be more promising in term of both skeletal (especially when associated with calcium supplementation) and extra-skeletal outcomes [105,106].

Alternatively, a cholecalciferol load dosage of 100,000–150,000 IU, followed by a maintenance dose of 2000 IU/day can be given [107].

In obese subjects, 25(OH)D levels are on average lower than in non-obese subjects, and often inversely correlated with body weight [104,108,109]. However, the relationship between serum 25(OH)D and body mass index (BMI) remains controversial, since negative, positive or absent correlations have been reported [110]. In general, the average reduction of 25(OH)D in subjects with increased BMI largely reflects the wider volume of distribution due to the increased fat mass [111], although other factors could also play a role [108]. After supplementation with vitamin D, the increase in 25(OH)D is reported to be inversely correlated with BMI [112]. Thus, in obese individuals, the cholecalciferol dosage could be increased by about 30%, compared to the dose used in individuals with normal BMI, in order to reach the same therapeutic efficacy [4,70,112,113,114,115,116]. A metanalysis of 18 RCTs performed in overweight/obese women (with cholecalciferol dosages ranging from 400 IU to 5714 IU per day) suggested that 1000 IU/daily vitamin D supplementation can suppress serum PTH levels, while 4000 IU/daily of vitamin D was associated with the largest increase in serum 25OHD levels [117].

Based on the clinical setting, a determination of 25(OH)D levels at 3–6 months from the start of treatment (depending on the dosage and the pro-hormone used) may be considered, possibly in the winter season (when the cutaneous synthesis of vitamin D is maximally impaired), for verifying the achievement of target values. If a loading dose of vitamin D is used, a verification of the levels of 25(OH)D can be performed after 1–2 months. In the absence of significant risk factors (i.e., weight changes, poor adherence to treatment, malabsorption syndromes or persistence of osteomalacia/myopathy symptoms) it is not recommended to repeat this determination once the desired levels have been obtained. In fact, there is very weak evidence that systematic, periodic 25(OH)D testing over time during supplementation may produce some benefit [26].

In conclusion, the optimal therapeutic regimen for vitamin D supplementation is currently unclear. However, much of the evidence in the literature suggests that, regardless of the dosing schedule (daily, weekly or monthly), the same results are obtained, providing that all potential factors affecting inter-individual variation are taken into consideration [5]. The only exception is the administration of mega-doses of vitamin D (>150,000 IU in a single bolus per year or every 6 months), due to the possibility of increasing the risk of falls [118], and, eventually, of fractures, as reported in a single study of patients without vitamin D deficiency [118]. Some other studies indicate that the use of large doses of vitamin D may be associated with acute increases in bone resorption markers which may explain the negative clinical results obtained by using intermittent high doses of vitamin D to treat or prevent vitamin D deficiency [119,120].

A possible alternative to the use of cholecalciferol is the use of its 25(OH)D metabolite, calcifediol. However, at present, the lack of evidence of anti-fracture efficacy, the limited half-life of 2–3 weeks, and the partial escape from physiological control of vitamin D metabolism should be considered. [15,17,19,33,34,41]. This metabolite, due to a better absorption (higher than that of cholecalciferol), a different volume of distribution, and the independence from hepatic hydroxylation, is able to quickly increase circulating 25(OH)D levels [92,93,121]. In a recent study, the administration of 20–40 mcg/day brought the levels of 25(OH)D above 30 ng/mL (75 nmol/L) after 20 days in all the subjects examined [94].

Importantly, in the case of vitamin D supplementation (both with cholecalciferol or calcifediol), an adequate intake of calcium with the diet (about 1000 mg/day) or, eventually, with calcium supplements (800–1000 mg/day) should be also guaranteed to optimize the benefits of vitamin D [122] (Table 6).

### 3.7. Should the General Population Be Supplemented with Vitamin D?

The rationale for a potential supplementation with vitamin D to all subjects in a primary prevention strategy arises from the following considerations: (a) wide prevalence of subjects with 25(OH)D below the “sufficient” threshold [123], (b) potential benefits on multiple (skeletal and extra-skeletal) outcomes [124], (c) safety profile of supplementation with cholecalciferol [4,125], and (d) simple and economic treatment [126].

However, does the administration of vitamin D to the general population reduce the risk of skeletal fractures or provide extra-skeletal benefits?

A meta-analysis was published in 2017 specifically including all RCTs conducted on non-hospitalized adult subjects [127]. In this setting, vitamin D supplementation did not change the risk for femoral fractures (RR 1.21; 95% CI 0.99–1.47) in 20,672 participants from 9 trials, nor for all fractures (RR 1.01; 95% CI 0.87–1.17) in 13,106 participants from 14 trials, versus placebo/no treatment. Likewise, the combination of vitamin D with calcium also did not significantly change the risk of femoral fractures (RR 1.09; 95% CI 0.85–1.39) in 17.927 participants from seven trials, nor of all fractures (RR 0.90; 95% CI 0.78–1.04) in 10,064 participants from eight trials. The authors concluded that available data did not support routine vitamin D treatment in non-hospitalized people for the prevention of fractures.

These conclusions were confirmed by a more recent meta-analysis focused on the general population (with the explicit exclusion of subjects hospitalized, or known to have vitamin D deficiency, osteoporosis or previous fractures), assessing specifically the fracture and mortality endpoints [34]. For the combined treatment with vitamin D and calcium, the analysis was limited to the WHI study [128], as only one other trial was available that reported a single fracture in the observation period [129]. Among 36,282 participants, the HR was 0.96 (95% CI 0.91–1.02) for all fractures and 0.88 (95% CI 0.72–1.08) for femur fractures. The authors concluded that there were no effects of vitamin D alone or in combination with calcium on the incidence of fractures in the general population. Moreover, no significant effect on all-cause mortality (RR 0.91; 95% CI 0.82–1.01) or incident cardiovascular disease emerged with vitamin D supplementation.

Previously, the effects of vitamin D supplementation on mortality had been investigated in four different meta-analyses published in 2014 [126,130,131,132]. Despite some elements of discordance and the use of different criteria for study inclusion, all four meta-analyses revealed a favorable effect of vitamin D on mortality, either in the main analysis or in some sub-analyses. However, all these meta-analyses were cautious with their conclusions, indicating that further studies were required before stating the efficacy of vitamin D in reducing mortality. Moreover, the population studied in these four meta-analyses mainly included post-menopausal women, mostly with advanced age.

The efficacy of vitamin D supplementation in the general healthy population on several extra-skeletal endpoints was judged to be poor and contradictory in a 2017 report by an ESCEO working group [25], as well as in a more recent and updated publication from an expert panel [133]. This is also consistent with the results of two meta-analyses published in 2018, which in turn did not support the efficacy of vitamin D on extra-skeletal endpoints. The first [134] included 13 RCTs (18,808 participants) and did not detect protective effects of vitamin D on tumor incidence and cancer mortality. The second included 31 RCTs (2751 participants) and did not reveal any protective effect of vitamin D supplementation on most of the considered vascular endpoints [135] (Table 7).

### 3.8. How Should the Patient with Chronic Kidney Disease (CKD) Be Supplemented with Vitamin D?

The reduction in circulating levels of 25(OH)D limits substrate availability for the synthesis of calcitriol in the kidney, thus aggravating the effects of the already reduced capacity of hydroxylation to 1,25(OH)2D and contributing to the development of secondary hyperparathyroidism. Indeed, serum 25(OH)D inversely correlates with PTH levels at all stages of CKD (126). In addition to common risk factors (i.e., sun exposure, food intake), hypovitaminosis D can be influenced by factors directly related to renal disease, such as loss of 25(OH)D and vitamin D binding protein in patients with proteinuria or hepatic hydroxylation defects of the vitamin D [136,137,138]. Supplementation with cholecalciferol, ergocalciferol or calcifediol at adequate doses is able to normalize the levels of 25(OH)D and reduce PTH levels in CKD [139,140,141]. Normalization of 25(OH)D may improve bone mineralization [142,143]. The effect of supplementation on hyperparathyroidism is variable, but in general, the correction of hypovitaminosis D can partially reduce PTH levels. However this is not observed in advanced stages of CKD and particularly in patients undergoing dialysis [139,140,141]. Evidence on the use of calcifediol in this setting is limited. The use of active vitamin D (calcitriol or synthetic analogues) should be restricted to patients on dialysis whose PTH values progressively increase [144].

As with the general population, the optimal levels of circulating 25(OH)D in the population with CKD are uncertain (136). The most recent KDIGO guidelines suggest correcting the vitamin D deficiency in the CKD in the same way as in the general population [144]. A multidisciplinary group of experts recommends the weekly administration of cholecalciferol, at the initial dose of 2000 IU/day with the aim of reaching 25(OH)D values of at least 30 ng/mL [145] (Table 8).

### 3.9. How Should the Patient with Chronic Liver Disease or under Treatment with Drugs Interfering with Hepatic Metabolism Be Supplemented with Vitamin D?

Insufficient levels of 25OHD are common in patients with chronic liver diseases, due to various mechanisms, including malnutrition, reduced sun exposure, malabsorption, reduced synthesis of binding proteins (i.e., DBP) and albumin. Furthermore, the hepatic metabolism of vitamin D can be compromised, with a reduced efficiency of 25-hydroxylation or an increase in calcifediol catabolism [146,147]. However, the importance of reduced hydroxylation is controversial and probably limited to the more advanced stages of liver failure [148]. Supplementation with cholecalciferol at a mean dosage of 2000 IU/day is effective in increasing 25(OH)D levels in a variety of chronic liver diseases (i.e., HCV-related hepatitis, non-alcoholic hepatic steatosis, cirrhosis), although the available evidence is not sufficient to show effects on morbidity or mortality [149]. The use of calcifediol can be considered as an alternative approach, even though evidence of efficacy remains limited [150,151]. Supplementation with vitamin D is necessary in case of concomitant therapies for osteoporosis. Many drugs interacting with Pregnane X receptor (PXR) stimulate the expression of 24-hydroxylases which increase the degradation of 25(OH)D, leading to vitamin D deficiency. The paradigmatic example is represented by the anti-convulsant inducers of cytochrome P450 (e.g., carbamazepine, phenobarbital, diphenylhydantoin) [152,153,154,155], but other drugs can also bind to PXR and interfere with the metabolism of vitamin D (i.e., glucocorticoids, anti-neoplastic agents, antiretrovirals, anti-tuberculous antibiotics) [153].

Supplementation with native vitamin D can compensate for these alterations and improve bone turnover indices and PTH levels, if the dosage is appropriate [156]. Supplementation is also necessary in case of concomitant therapies for osteoporosis. In severe liver diseases, the use of cholecalciferol at the same dosages than in osteoporotic patients [157,158] has been suggested. The use of calcifediol may be considered as an alternative approach [157] (Table 9).

### 3.10. In the Subject Who Need Supplementation, Might Vitamin D Have Toxic Effects?

“Classic” vitamin D intoxication manifestations include hypercalcemia and hypercalciuria. These events are to be considered exceptional with the administration of cholecalciferol or ergocalciferol and may occur only in case of particularly high dosages, with circulating calcifediol levels around 150–200 ng/mL (375–500 nmol/L) or above [4,132]. Conversely, these manifestations may occur more frequently with the recommended dosages of calcitriol or alfa-calcidiol. A Cochrane meta-analysis [159], based on 710 subjects included in three trials, calculated an RR of hypercalcemia of 3.18 (CI 95% 1.17–8.68). The excess of absolute risk of hypercalcemia with calcitriol or alfa-calcidiol was 5.55%, although it should be emphasized that these were mostly non-serious forms. In the same meta-analysis, the intake of calcium supplements together with non-hydroxylated vitamin D showed a significant increase in the risk of nephrolithiasis (RR 1.17; 95% CI 1.02–1.34), although the risk excess was modest (0.33%). It should be noted, however, that data concerning nephrolithiasis was derived almost exclusively from the WHI study, which contributed to over 99% of subjects included in the meta-analysis. In that study, patients had unlimited access to alimentary calcium [128]. Thus, given the characteristics of the subjects included in the WHI study, this information may not be applicable to populations, such as the Italian one, characterized by a low dietary calcium intake and which do not make common use of calcium supplements or fortified foods.

An increased risk of falling was described with: (a) 100,000 IU cholecalciferol per month towards lower doses in nursing home residents [160]; (b) 500,000 IU cholecalciferol once a year *vs.* placebo in elderly women at high risk of falls/fracture not admitted to an institution [118]; (c) 60,000 IU of cholecalciferol or 24,000 IU of cholecalciferol and 12,000 IU of calcifediol per month towards a lower dose of cholecalciferol in elderly (>70 yrs), non-institutionalized men and women with a personal history of fall [130]; and (d) 4000 IU cholecalciferol per day to smaller doses in women aged >57 years and basal calcifediol values ≤ 20 ng/mL (50 nmol/L) [161].

However these data relating vitamin D supplementation to an increase in the risk of falls do not refer to the general population, but to elderly subjects at high risk of falling and mostly not deficient in vitamin D (excluding the small trial by Smith et al. [161]) at the start of treatment. A very recent meta-analysis [162] reported contradictory results on the risk of falling in different trials with no significant overall effect on subjects not admitted to nursing homes. In contrast, other meta-analyses supported a favorable, preventive role of vitamin D on the risk of falling [163,164], which was prevalent in subjects who were vitamin D deficient at baseline and treated with dosages able to increase circulating 25(OH)D levels around 30 ng/mL (75 nmol/L) [163,164,165].

## 4. Conclusions

This position statement is primarily intended for use by clinicians in order to face the issues of the definition, assessment, and management of vitamin D inadequacy, in order to (a) improve and standardize the “clinical practice”; (b) offer the patient the indications for “best care”, to be followed uniformly at national level; and (c) guarantee an evidence-based reference for national and regional institutions, for regulatory organizations and payers.

Given these premises, unavoidably, the present work has some limitations. Firstly, this document could not be considered a systematic review of original studies since it is based even on the already existing guidelines on these topics. However, in the present work, we can confirm even in the Italian population what has been expressed by other guidelines previously published in the literature. In addition, at variance with most previous guidelines, we have applied the “P.I.C.O.” criterion and the GRADE system, which represent a systematic approach for rating the certainty of evidence in systematic reviews and other evidence syntheses, implying, in fact, a comprehensive systematic review.

Second, we decided not to face the problem of which has to be considered the normal vitamin D levels. At variance, we propose a personalized approach for the use of vitamin D supplementation. As suggested by IOM [166], we propose that in the general population, including healthy elderly individuals, a threshold value of 25(OH)D ≥ 20 ng/mL (50 nmol/L) should be considered as adequate and should not require any supplementation. Conversely, in patients with osteoporosis or other disorders of bone metabolism—especially when treated with bone modifying agents—as well as in subjects at risk for hypovitaminosis D, a value of ≥30 ng/mL can be suggested as “optimal”.

The recent paper by LeBoff and coauthors reinforces the idea that vitamin D supplementation should not be provided to subjects without hypovitaminosis D or osteoporosis. The authors found that in subjects without hypovitaminosis D (87% of subjects had vitamin D values greater than 20 ng/mL) and not recruited for low bone mass and/or osteoporosis, the vitamin D supplementation was not effective in reducing the risk of fractures. These data, therefore, cannot question the positive skeletal effects of vitamin D supplementation (in association with calcium supplementation where necessary) in subjects with hypovitaminosis D, with low bone mass or osteoporosis, but reinforce the idea that vitamin D supplementation should not be given in an indiscriminate way [167].

We are aware that multiple large interventional trials show no benefit other than in upper respiratory tract infections of vitamin D supplementation on fracture risk. However, it should be also considered that many of these trials suffered for one or more among the following issues: (1) high-risk patients not included; (2) low number of patients with inadequate 25(OH)D levels; (3) lack of adequate dose of vitamin D supplementation and/or lack of adjustment for inadequate dietary calcium intake; (4) lack of adequate 25(OH)D concentration during the whole study duration; (5) a study duration not sufficient for a reliable evaluation of BMD changes and fracture incidence; (6) the lack of the registration of all major comorbidities [168].

A further limitation of the present Position Statement is that we decided not to face the problems of the need for vitamin D supplementation as related to the prevention of autoimmunity and in patients with inflammatory bowel disorders. Recent data seem to suggest a possible link between hypovitaminosis D and both autoimmunity [169,170] and inflammatory bowel disease [171,172]. Notwithstanding these recent data, our decision has been based on the fact that for the use of vitamin D in these disorders, data in the literature, in terms of large controlled and randomized intervention clinical trials, are lacking.

In addition, even the issue regarding the need for supplementation with vitamin D analogs in the case of chronic renal failure (CKD) has been not included among the aims of the present Position Statement, since this topic would have required a separate document. The readers can refer to a nice review by Brandenburg and Ketteler summarizing the developments of vitamin D therapies in CKD patients of the last decades and individuating crucial issues for future research in particular on the optimal PTH level for CKD patients and on the optimal vitamin D level to support optimal PTH titration [173]. Nowadays, the Experts Panel believe that the K-DIGO Guidelines should be considered as the standard for the vitamin D supplementation in subjects with renal failure [173].

Finally, we could not give evidence-based suggestions for how long vitamin D supplementation should be maintained. No data are available regarding whether or not vitamin D supplementation should be administered indefinitely. However, it seems advisable that in patients with hypovitaminosis D, the vitamin D supplementation with a regimen able to maintain “optimal” vitamin D levels should be provided until the cause of vitamin D deficiency has been removed. Similarly, in patients with osteoporosis, vitamin D levels should be maintained adequate for the duration of the anti-osteoporotic therapy. Studies focusing the potential barriers to treatment and how to overcome them in patients with fragility fractures and/or with risk factors for osteoporosis are warranted.

## Figures and Tables

**Table 1 nutrients-14-04148-t001:** Definition of Vitamin D Status.

	Deficiency *	Insufficiency *	Optimal * Optimum *
GENERAL POPULATION	<10 ng/mL	<20 ng/mL	**20–50 ng/mL**
POPULATION AT RISK ** OR ON TREATMENT WITH *BONE MODIFYING AGENTS*	<10 ng/mL	<30 ng/mL	**30–50 ng/mL**

* Reported cut-off values should be considered with a margin of variability of +/−10%, considering the analytical variability of the 25(OH)D dosage. Moreover, due to the seasonal variability of 25(OH)D levels, a dosage performed at the end of winter/early spring should be particularly considered. A serum value of <10 ng/mL (25 nmol/L) is associated with rickets and osteomalacia, if long lasting. From ng/mL to nmol/L: ng/mL × 2.5. ** The population at risk of hypovitaminosis is shown in Table 2.

**Table 3 nutrients-14-04148-t003:** Recommendation, and its evidence level, for not to perform 25(OH)D circulating levels measurement.

	Evidence Level
**It is recommended** not to perform the 25(OH)D measurement in the general population.	**⊕**

**Table 4 nutrients-14-04148-t004:** Evidence levels supporting the suggestion and recommendation regarding the measurements of 25(OH)D levels in specific categories of subjects.

	Evidence Levels
**It is suggested not to** indiscriminately measure the levels of 25(OH)D in patients with conditions/pathologies at risk of hypovitaminosis D	**⊕⊕**
**It is recommended** the measurement of 25(OH)D levels only when it is deemed necessary for the clinical management of the patient (i.e., when osteomalacia is suspected)	**⊕⊕**

**Table 5 nutrients-14-04148-t005:** Evidence levels in support of the suggestion not to carry out the measurement of serum values of 25 (OH) D in the specific categories of subjects/patients described here.

	Evidence Levels
**We suggest** that baseline levels of 25(OH)D should not be routinely assessed in patient candidates for pharmacological treatment for osteoporosis or other metabolic bone disorders (that are mandatorily associated with vitamin D supplementation), unless osteomalacia is suspected.	**⊕⊕**

**Table 6 nutrients-14-04148-t006:** Suggestions and recommendations concerning vitamin D supplementation in subjects with hypovitaminosis D or candidates to receive anti-fracture drugs.

In Subjects with Hypovitaminosis D, or Candidates for Bone Active Agents for Osteoporosis:	Evidence Levels
**We suggest** a dose of cholecalciferol supplementation between 800 IU/day and 2000 IU/day. There is no single, fixed dose for all subjects that needs to be supplemented.	**⊕**
**We suggest** a daily, weekly, monthly schedule based on the dose administered. In these settings, the maximum single daily dose to be administered should not exceed 100,000 IU. An adequate calcium intake (800–1000 mg/day) must always be ensured.	**⊕**
**We recommend** the use of an initial loading dose, followed by the maintenance dose in patients with symptomatic osteomalacia and/or serum 25(OH)D < 10 ng/mL, or in patients starting bone anti-resorptive therapy with intravenous bisphosphonates or denosumab with serum 25(OH)D < 20 ng/mL.	**⊕⊕⊕**
**We recommend**, as loading dose, cholecalciferol 3000–10,000 IU/day (average 5000 IU/day) for 1–2 months, or cholecalciferol in a single dose of 60,000 to 150,000 IU followed by the maintenance dose (2000 IU/day). Alternatively, we suggested calcifediol 20–40 mcg/day (4–8 gtt/day) for 20–30 days, before switching to maintenance dose *.	**⊕⊕⊕**

* With a limited recommendation for a faster normalization of serum levels of 25(OH)D only.

**Table 7 nutrients-14-04148-t007:** No evidence-based conclusions can currently be drawn on potential benefits of vitamin D in the general population, both in terms of cost-effectiveness and in terms of mortality or on skeletal and extra-skeletal outcomes.

	Evidence Levels
**It is recommended** not to administer vitamin D supplements in the general population, since there is no definite evidence of cost-effective benefits, either on mortality or on skeletal and extra-skeletal outcomes.	**⊕⊕⊕**

**Table 8 nutrients-14-04148-t008:** Recommendations regarding the supplementation of vitamin D metabolites in patients with impaired renal function, and in relation to their stage of renal failure.

	Evidence Level
**It is recommended** in the patient with CKD-MBD to correct hypovitaminosis D with cholecalciferol, with the same modalities used in the general population with normal renal function.	**⊕⊕⊕⊕**
**It is recommended** to limit the use of active vitamin D compounds (calcitriol or synthetic analogues) to subjects on dialysis or in G4-G5 CKD stage with severe and progressive hyperparathyroidism	**⊕⊕⊕⊕**

**Table 9 nutrients-14-04148-t009:** Suggestions regarding the supplementation of vitamin D metabolites in subjects suffering from severe hepatic insufficiency or undergoing therapies that interfere with the hepatic metabolism of vitamin D.

	Evidence Level
**We suggest** supplementation with at least 2000 IU/day of cholecalciferol in patients with severe hepatic insufficiency or in case of chronic therapies that interfere with the hepatic vitamin D metabolism. The use of calcifediol is a possible alternative.	**⊕**

## Data Availability

This study did not report any data.

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
