# Peer review of "Definition, Assessment, and Management of Vitamin D Inadequacy: Suggestions, Recommendations, and Warnings from the Italian Society for Osteoporosis, Mineral Metabolism and Bone Diseases (SIOMMMS)"

_nutrients, 2022, doi:10.3390/nu14194148_

Round 1

Reviewer 1 Report

A review by

Bertoldo et al. aimed to review 10 questions on vitamin D around vitamin D intake level, 25(OH)D levels regarding osteoporosis, and vitamin D supplements’ effectiveness on health outcomes. While these questions are important to ask, as there is mixed evidence with different views on the “suitable” or “effective” amount, I find the answer to each question is primarily based on what has been defined in previous guidelines or results of single studies rather than conducting an updated comprehensive systematic review to reach the recommendations. Additionally, the authors defined the cut-off for vitamin D’s adequate level as incorrect based on the IOM report. Please see what the committee argued against the level set by the Endocrine society here, https://www.ncbi.nlm.nih.gov/pmc/articles/PMC5393439/#FN1.

 The authors should also know that a  well-designed RCT of over 25,000 multi-ethnic US participants was published in NEJM in July 2022. The study found no risk reduction of 2000 IU / day of vitamin D supplements on fractures, in different strata of baseline serum vitamin D level, from 12ng/mL ~ 37ng/mL, by osteoporosis medication use, taking vitamin D / calcium supplement, fracture history, age, sex at baseline. 

Author Response

Reviewer #1

A review by Bertoldo et al. aimed to review 10 questions on vitamin D around vitamin D intake level, 25(OH)D levels regarding osteoporosis, and vitamin D supplements’ effectiveness on health outcomes. While these questions are important to ask, as there is mixed evidence with different views on the “suitable” or “effective” amount, I find the answer to each question is primarily based on what has been defined in previous guidelines or results of single studies rather than conducting an updated comprehensive systematic review to reach the recommendations.

Additionally, the authors defined the cut-off for vitamin D’s adequate level as incorrect based on the IOM report. Please see what the committee argued against the level set by the Endocrine society here, https://www.ncbi.nlm.nih.gov/pmc/articles/PMC5393439/#FN1. The authors should also know that a well-designed RCT of over 25,000 multi-ethnic US participants was published in NEJM in July 2022. The study found no risk reduction of 2000 IU / day of vitamin D supplements on fractures, in different strata of baseline serum vitamin D level, from 12ng/mL ~ 37ng/mL, by osteoporosis medication use, taking vitamin D/calcium supplement, fracture history, age, sex at baseline.

Question 1: “I find the answer to each question is primarily based on what has been defined in previous guidelines or results of single studies rather than conducting an updated comprehensive systematic review…”.

Answer 1: The authors thank the reviewer for suggesting improvement points regarding the submitted paper. Indeed, in the present work we can confirm even in the Italian population what has been expressed, more or less clearly, by other guidelines previously published in the literature, indirectly identifying the good work done in full by the drafters of the documents produced, including ours. However, we would like to underline that not all the previous guidelines have applied the "P.I.C.O." criterion, as done by our panel of experts. Additionally, in the present work, differently from the previous publications, the GRADE system has been applied, which is a systematic approach for rating the certainty of evidence in systematic reviews and other evidence syntheses, implying, in fact, a comprehensive systematic review. This is now reported on page 15 (lines 24-39) in the revised version of the manuscript

Question 2: “what the committee argued against the level set by the Endocrine society here, https://www.ncbi.nlm.nih.gov/pmc/articles/PMC5393439/#FN1”

Answers 2: We agree that this is an important point. Indeed, as suggested by the IOM committee, identifying a personalized approach for the use of vitamin D supplementation is the general aim of the present work. Indeed, as suggested by IOM we do not believe that all persons with serum 25OHD levels below 20 ng/ml are deficient in vitamin D and should be supplemented, but that the criterion for vitamin D supplementation depends on which patient we are leading with. Accordingly, the present experts panel propose that in the general population, including healthy elderly individuals, a threshold value of 25(OH)D ≥ 20 ng/ml (50 nmol/l) should be considered as adequate and should not require any supplementation. Conversely, in patients with osteoporosis or other disorders of bone metabolism, especially when treated with bone modifying agents, as well as in subjects at risk for hypovitaminosis D, a value of ≥ 30 ng/ml can be suggested as “optimal”. This is stated on page 6 (last paragraph) and this point is discussed on page 15 (lines 25-34) of the manuscript. However, this panel is keen to point out that IOM indications are questionable especially in countries, such as Italy, where there is no use of vitamin D fortified foods.

Question 3: “The authors should also know that a well-designed RCT of over 25,000 multi-ethnic US participants was published in NEJM in July 2022. The study found no risk reduction of 2000 IU / day of vitamin D supplements on fractures, in different strata of baseline serum vitamin D level, from 12ng/mL ~ 37ng/mL, by osteoporosis medication use, taking vitamin D / calcium supplement, fracture history, age, sex at baseline.”

Answer 3: We thank the reviewer for the suggestion. The very recent paper by LeBoff and coauthors has been included within text and mentioned in references. In fact, as correctly stated by the authors themselves, this study simply reinforces the idea that the vitamin D supplementation should not be provided to subjects without hypovitaminosis D or osteoporosis. Indeed, the authors found that in subject without hypovitaminosis D (87% of subjects had vitamin D values greater than 20 ng/ml) and not recruited for low bone mass and/or osteoporosis (only about 10% of subjects with history of fragility fracture), the vitamin D supplementation was not effective in reducing the risk of fractures. In particular, only 400 subjects had severe vitamin D deficiency and no data regarding their bone health are provided. Importantly, the study evaluated the effect of vitamin D supplementation on all kind of fracture and not only the fragility ones. These data, therefore, cannot question the positive skeletal effects of vitamin D supplementation (in association with calcium supplementation where necessary) in subjects with hypovitaminosis D, with low bone mass or osteoporosis, but reinforce the idea that vitamin D supplementation should not be given in an indiscriminate way. This is now reported on page 15 (lines 34-43) in the new version of the manuscript. However, this panel considers at least imprecise defining LeBoff's study as “well designed” since vitamin D is given to those who do not need it.

Reviewer 2 Report

Based on the above premises, and in the light of the new evidences from the literature, the SIOMMMS board felt the necessity to revise and update its original recommendations about the definition, prevention and treatment of vitamin D deficiency in adults,  released in 2011. 

The authors raise 10 questions that are key in approaching vitamin D supplementation

The authors analyze the issues raised based on meta-analyses that constitute the highest levels of evidence. They establish a gradation in their recommendations In those aspects where the evidence does not demonstrate the effectiveness of the measures taken, they emphasize it. The guideline is useful for clinical practice, is sound based on the published literature, and it´is well written.

Author Response

Reviewer #2

Based on the above premises, and in the light of the new evidences from the literature, the SIOMMMS board felt the necessity to revise and update its original recommendations about the definition, prevention and treatment of vitamin D deficiency in adults, released in 2011. The authors raise 10 questions that are key in approaching vitamin D supplementation. The authors analyze the issues raised based on meta-analyses that constitute the highest levels of evidence. They establish a gradation in their recommendations in those aspects where the evidence does not demonstrate the effectiveness of the measures taken, they emphasize it. The guideline is useful for clinical practice, is sound based on the published literature, and it’s well written.

Answer: the authors sincerely thank the reviewer for having fully understood the essence of the spirit that motivated the need for S.I.O.M.M.M.S. to write this document in this way, which, moreover, was done with full intellectual honesty and expository "humility".

Reviewer 3 Report

Thank you for the opportunity to read this manuscript on vitamin D and various issues related to vitamin D status/assessment/supplementation.

Unfortunately I do not feel this manuscript adds new or useful information to the field, and possibly just adds to misinformation beginning with the very first sentence

"It has been well established that Vitamin D deficiency is a world wide concern." This is not a well established fact!

Vitamin D deficiency is characterised by either osteomalacia or rickets - and we do not have an epidemic of either. There is evidence for vitamin D blood levels that place people at increased risk of deficiency - and as the authors write there is controversy about how many people we are talking about as there is little agreement about "normal" blood levels. Risk of deficiency is not the same as deficiency.

The authors write that "optimal" levels of vitamin 25(OH)vitD should be 125nmol/L -this is not supported by the evidence. In any population! 

The authors should review the recently released ancillary Vital study 

"Supplemental Vitamin D and Incident Fractures in Midlife and Older Adults" (NEJM, July 2022)

in which "LeBoff and colleagues report that, contrary to expectations, vitamin D3 did not reduce the risk of fractures over a median follow-up of 5.3 years, even in the 20% of the participants taking supplemental calcium at a dose of up to 1200 mg per day.""The fact that vitamin D had no effect on fractures should put to rest any notion of an important benefit of vitamin D alone to prevent fractures in the larger population"

This study also  included the 400 participants with levels below 12 ng/L, and also patients with fragility and/or on anti-osteoporotic fracture medicines.

This study is just one of many clearing demonstrating that vitamin D supplementation does not benefit people re fracture risk - at least with levels that most scientists agree are "normal" and even low normal.

Author Response

Reviewer #3

Thank you for the opportunity to read this manuscript on vitamin D and various issues related to vitamin D status/assessment/supplementation. Unfortunately, I do not feel this manuscript adds new or useful information to the field, and possibly just adds to misinformation beginning with the very first sentence. "It has been well established that Vitamin D deficiency is a worldwide concern." This is not a well-established fact! Vitamin D deficiency is characterized by either osteomalacia or rickets - and we do not have an epidemic of either. There is evidence for vitamin D blood levels that place people at increased risk of deficiency - and as the authors write there is controversy about how many people we are talking about as there is little agreement about "normal" blood levels. Risk of deficiency is not the same as deficiency. The authors write that "optimal" levels of vitamin 25(OH)vitD should be 125nmol/L -this is not supported by the evidence. In any population! The authors should review the recently released ancillary Vital study "Supplemental Vitamin D and Incident Fractures in Midlife and Older Adults" (NEJM, July 2022) in which "LeBoff and colleagues report that, contrary to expectations, vitamin D3 did not reduce the risk of fractures over a median follow-up of 5.3 years, even in the 20% of the participants taking supplemental calcium at a dose of up to 1200 mg per day." "The fact that vitamin D had no effect on fractures should put to rest any notion of an important benefit of vitamin D alone to prevent fractures in the larger population". This study also included the 400 participants with levels below 12 ng/L, and also patients with fragility and/or on anti-osteoporotic fracture medicines. This study is just one of many clearings demonstrating that vitamin D supplementation does not benefit people re fracture risk - at least with levels that most scientists agree are "normal" and even low normal.

Question 1: "It has been well established that Vitamin D deficiency is a worldwide concern." This is not a well-established fact!”

Answer 1: The authors feel obliged to thank this reviewer very much for having raised important issues that undoubtedly need to be enlightened. The concept that vitamin D deficiency is or is not a worldwide problem is somewhat questionable, both in the affirmative and in the negative sense. The international literature has, in fact, endowed us with numerous works that consider it a worldwide problem. An academic distinction could be made, but perhaps with no clear significant use, if we have to consider the clinical presence of “worldwide” associated rickets/osteomalacia, especially when not exhibiting an overt phenotype expression, or consider only the “biochemical” sign of inadequate circulating values of 25OHD. It is well established that severe vitamin D deficiency causes rickets and osteomalacia, both representing skeletal diseases with potential health and economic burden. One of the major aims of supplementing vitamin D (particularly in those nations without a food fortification campaign) is indeed the prevention of both disorders so that they could not became again a relevant health related issue. We have now modified with “severe vitamin D deficiency”. However, we believe that the existence of pandemic insufficiency/ deficiency, both in biochemical and osteomalacic terms (our paper does not touch on pediatric aspects) in specific population groups is hardly questionable, such as the elderly one, institutionalized or not, as widely report in literature.

Question 2: “There is evidence for vitamin D blood levels that place people at increased risk of deficiency - and as the authors write there is controversy about how many people we are talking about as there is little agreement about "normal" blood levels. Risk of deficiency is not the same as deficiency. The authors write that "optimal" levels of vitamin 25(OH)vit. D should be 125 nmol/L -this is not supported by the evidence. In any population!”

Answer 2:

The optimal levels set between 75 nmol/l and 125 nmol/L refer to the previous SIOMMMS Guidelines published in 2012.In accordance with the suggestions of the other reviewers, in the revised version of the present manuscript we more clearly state that in the general population, including healthy elderly individuals, a threshold value of 25(OH)D ≥ 20 ng/ml (50 nmol/l) should be considered as adequate and should not require any supplementation. Conversely, in patients with osteoporosis or other disorders of bone metabolism, especially when treated with bone modifying agents, as well as in subjects at risk for hypovitaminosis D, a value of ≥ 30 ng/ml can be suggested as “optimal”. This is stated on page 6 (last paragraph) and this point is discussed on page 15 (lines 25-34) of the manuscript

Question 3: “The authors should review the recently released ancillary Vital study "Supplemental Vitamin D and Incident Fractures in Midlife and Older Adults" (NEJM, July 2022) in which "LeBoff and colleagues report that, contrary to expectations, vitamin D3 did not reduce the risk of fractures over a median follow-up of 5.3 years, even in the 20% of the participants taking supplemental calcium at a dose of up to 1200 mg per day." "The fact that vitamin D had no effect on fractures should put to rest any notion of an important benefit of vitamin D alone to prevent fractures in the larger population". This study also included the 400 participants with levels below 12 ng/L, and also patients with fragility and/or on anti-osteoporotic fracture medicines. This study is just one of many clearings demonstrating that vitamin D supplementation does not benefit people re fracture risk - at least with levels that most scientists agree are "normal" and even low normal.”

Answer 3: The panel believes it should be clarified that the paper by LeBoff and colleagues was not yet available at the time of the "manuscript drafting". We thank anyway the reviewer for the suggestion. The very recent paper by LeBoff and coauthors has been included within text and mentioned in references. In fact, as correctly stated by the authors themselves, this study simply reinforces the idea that the vitamin D supplementation should not be provided to subjects without hypovitaminosis D or osteoporosis. Indeed, the authors found that in subject without hypovitaminosis D (87% of subjects had vitamin D values greater than 20 ng/ml) and not recruited for low bone mass and/or osteoporosis (only about 10% of subjects with history of fragility fracture), the vitamin D supplementation was not effective in reducing the risk of fractures. In particular, only 400 subjects had severe vitamin D deficiency and no data regarding their bone health were provided. Importantly, the study evaluated the effect of vitamin D supplementation on all kind of fracture and not only the fragility ones. These data, therefore, cannot question the positive skeletal effects of vitamin D supplementation (in association with calcium supplementation where necessary) in subjects with hypovitaminosis D, with low bone mass or osteoporosis, but reinforce the idea that vitamin D supplementation should not be given in an indiscriminate way. This is now discussed on page 15 (lines 34-43) in the new version of the manuscript.

Reviewer 4 Report

This is a very interesting and important paper on the need for supplementation with vitamin D for the prevention and treatment of osteoporosis.  

The authors need to state on the time limit of supplementation with vitamin D in subjects for the prevention and treatment of osteoporosis.  In other words, if indicated, vitamin D should or should not be administered indefinitely?

The authors should make a note on the need for vitamin D supplementation as related to the prevention of autoimmunity.  

The authors should make a note on the need for supplementation with vitamin D analogs in the case of renal failure.  

The authors should refer to the need of vitamin D supplementation in patients with inflammatory bowel disorders.  

The use of the English language should be improved.

Author Response

Reviewer #4

This is a very interesting and important paper on the need for supplementation with vitamin D for the prevention and treatment of osteoporosis. The authors need to state on the time limit of supplementation with vitamin D in subjects for the prevention and treatment of osteoporosis. In other words, if indicated, vitamin D should or should not be administered indefinitely? The authors should make a note on the need for vitamin D supplementation as related to the prevention of autoimmunity. The authors should make a note on the need for supplementation with vitamin D analogs in the case of renal failure. The authors should refer to the need of vitamin D supplementation in patients with inflammatory bowel disorders. The use of the English language should be improved.

Question 1 “…if indicated, vitamin D should or should not be administered indefinitely?”

Answer 1: We thank the reviewer for the suggestion. Indeed, we could not give evidence-based suggestions on for how long vitamin D supplementation should be maintained. Indeed, no data are available regarding whether or not vitamin D supplementation should be administered indefinitely. However, it seems advisable that in patients with hypovitaminosis D, the vitamin D supple-mentation with a regimen able to maintain “optimal” vitamin D levels should be pro-vided until the cause of vitamin D deficiency has been removed. Similarly, in patients with osteoporosis vitamin D levels should be maintained adequate for the duration of the anti-osteoporotic therapy. This point is now discussed on page 16 (lines 18-25) of the new version of the manuscript.

Question 2: “The authors should make a note on the need for vitamin D supplementation as related to the prevention of autoimmunity. The authors should refer to the need of vitamin D supplementation in patients with inflammatory bowel disorders.”

Answer 2: the authors thank this reviewer for raising important issues. However, the panel specifies that the submitted document is strictly inherent to the skeletal effects of vitamin D. For the use of vitamin D in autoimmune diseases, the data in the literature, in terms of large controlled and randomized intervention clinical trials, are lacking. Even the meta-analyzes produced have not so far shown univocal and shared results. The same applies to the use of vitamin D metabolites in inflammatory intestinal diseases. For all this we have chosen not to deal with these specific aspects. This point has been now reported as a study limitation (page 16, lines 1-8) and some updated literature has been included (ref# 177-180)

Question 3: The authors should make a note on the need for supplementation with vitamin D analogs in the case of renal failure

Answer 3: We agree that this is an important topic. However, the Expert Panel decided not to include the issue regarding the need for supplementation with vitamin D analogs in the case of renal failure among the aims of the present Position Statement, since this topic would have required a separate document, to be fully elucidated. The readers can refer to a nice review by Brandenburg & Ketteler summarizing the developments of vitamin D therapies in CKD patients of the last decades and individuating crucial issues for future research in particular on the optimal PTH level for CKD patients and on the optimal vitamin D level to support optimal PTH titration (ref #181). Nowadays, the Experts Panel believe that the K-DIGO Guidelines should be considered as the standard for the vitamin D supplementation in subjects with renal failure (ref # 182). This is now reported on page 16 (lines 9-17) of the new version of the manuscript

Question 4: The use of the English language should be improved.

Answer 4: the manuscript has been revised by a mother-tongue.

Round 2

Reviewer 3 Report

Thank you for the opportunity to re-read this manuscript.

Unfortunately, my original recommendation stands.

There are many assumptions made in this manuscript, often prefaced by “it is well known” - a sign, one of my own professors used to say, that it is most certainly not well known.

I base my recommendation on two substantial concerns:

1) The repeated statement that vitamin D deficiency is highly prevalent - I think this is incorrect and furthermore, the more accurate statement would  be, “at risk of deficiency”.

2) The authors’ position that 125 nmol/L vit D 25(OH) is the optimal blood level -  for which there is no robust evidence. Furthermore, I think this is a misleading and potentially dangerous reference range.

As the authors state testing of vitamin D blood levels is highly prevalent as is use of supplements - yet multiple interventional trials with thousands and thousands of patients show no benefit other than in upper respiratory tract infections and then with the recommended doses of 400-800 IU not higher.

This study, in my opinion, does not contribute new or useful information to the field. 

The authors state that there are patients with fragility fractures, risk factors for osteoporosis etc  not being treated - why not do research on the barriers to treatment and how to overcome them - certainly a topic under researched compared with vitamin D.

Author Response

Unfortunately, my original recommendation stands.

There are many assumptions made in this manuscript, often prefaced by “it is well known” - a sign, one of my own professors used to say, that it is most certainly not well known.

We have checked the manuscript and the phrase has been modified (page 1, abstract section)

I base my recommendation on two substantial concerns:

1) The repeated statement that vitamin D deficiency is highly prevalent - I think this is incorrect and furthermore, the more accurate statement would  be, “at risk of deficiency”.

We have modified the text in accordance with the reviewer (abstract lines 3-4, Introduction 5th paragraph)

2) The authors’ position that 125 nmol/L vit D 25(OH) is the optimal blood level -  for which there is no robust evidence. Furthermore, I think this is a misleading and potentially dangerous reference range.

In the present manuscript we refer to already published data. Anyway, in accordance with the reviewer suggestion, the text has been modified

As the authors state testing of vitamin D blood levels is highly prevalent as is use of supplements - yet multiple interventional trials with thousands and thousands of patients show no benefit other than in upper respiratory tract infections and then with the recommended doses of 400-800 IU not higher.

We are fully aware of the concerns raised by the reviewer. Consequently, in the new version of the manuscript this point has been discussed and a new reference has been added (Chiodini I, Gennari L. Falls, fractures and vitamin D: a never-ending story? Nat Rev Rheumatol 2019 Jan;15(1):6-8) (page 16, first paragraph)

This study, in my opinion, does not contribute new or useful information to the field. 

This position statement was primarily intended for use by clinicians in order to face the issues of the definition, assessment, and management of vitamin D inadequacy, in order to: a) improve and standardize the "clinical practice"; b) offer the patient the indications for "best care", to be followed uniformly at national level; c) guarantee an evidence-based reference for national and regional institutions, for regulatory organizations and payers, as stated in the Conclusions section (1st paragraph). Consequently, this paper was not intended to provide new information to the field, rather to give precise indications and recommendation for the definition, assessment, and management of vitamin D inadequacy

The authors state that there are patients with fragility fractures, risk factors for osteoporosis etc not being treated - why not do research on the barriers to treatment and how to overcome them - certainly a topic under researched compared with vitamin D.

This is a good point for future research. This has been now added in the Conclusions section (last sentence).